# Effect of Emulsion Particle Size on the Encapsulation Behavior and Oxidative Stability of Spray Microencapsulated Sweet Orange Oil (*Citrus aurantium* var. *dulcis*)

**DOI:** 10.3390/foods12010116

**Published:** 2022-12-26

**Authors:** Qun Peng, Ziyi Meng, Ziyang Luo, Hanying Duan, Hosahalli S. Ramaswamy, Chao Wang

**Affiliations:** 1Department of Food Science and Technology, Jinan University, Guangzhou 510632, China; 2Department of Food Science and Agricultural Chemistry, Macdonald Campus of McGill University, Montréal, QC H3A 0G4, Canada

**Keywords:** particle size, water activity, release characteristic, oxidative stability, powder morphology

## Abstract

Three different feed emulsions of different particle sizes were mixed with a modified starch and maltodextrin and spray dried to make a large (LP), small (SP), and nano-size encapsulated powder (NP), respectively. Emulsion size, oil content, loading capacity (LC), encapsulation efficiency (EE), water content, a_w_, scanning electron microscopy (SEM), glass transition temperature (*Tg*), as well as d-limonene release characteristic and limonene oxide formation rate during 37 °C and various a_w_ storage were determined. With the increase of the feed emulsion size, the reconstituted emulsion size of the LP tended to increase and change to a bimodal distribution. The surface oil content increased with the increasing size of the reconstituted emulsion, and the opposite was true for EE. The smaller the reconstituted emulsion size, the higher *Tg* during a low a_w_ condition. The *Tg* of the LP, SP and NP were 62, 88, and 100 °C, respectively, and NP > SP > LP. The release and the oxidative rate of d-limonene was the lowest for the NP and then increased for the SP and LP. The release and oxidative rates increased with the elevation of a_w_ and peaked at 0.33. The powder surface morphological structure was intact, the spray-dried powder was more stable, and microstructure changed from a glass state to a rubbery state during storage.

## 1. Introduction

Sweet orange (*Citrus aurantium* var. *dulcis*) essential oil (SOEO) is attractive as it possesses fresh orange flavor, antioxidant activity, antibacterial ability, and other functions. The predominant active compound in SOEO is d-limonene [1]. Due to the oil base, high sensitivity, and strong volatility of SOEO, the major loss and oxidation of d-limonene have been reported during processing, storage, and transportation [2]. The cut-back and freezing concentration processes adopted for orange juice concentrate are primarily due to the thermal sensitivity of orange flavors during the evaporation process [3]. For the same reason, to prepare commercial orange juice from concentrated orange juice, juice with high solids content is diluted with reconstituted SOEO flavor [4]. In order to reduce the oxidation and to control the release of d-limonene, a microencapsulation process was exploited [5]. Some studies have evaluated this retention, oxidative stability, and release of flavor during production and storage. However, the fundamental question remaining is how exactly the particle size of spray-dried encapsulation correlates with the stability and release of the flavor compounds [6].

The oxidation and release of spray-dried flavor compounds depends on the following factors: the carrier material [7], the type of atomization and processing temperature [8], the emulsion size [9], the powder particle size [10], the powder morphology [11] and the water activity. Reineccius and Yan [12] and Subramaniam et al. [13] found that particle properties, including wall thickness and particle surface composition, are very important because they may affect the exposure of the outer surface to oxygen and the diffusion of the oxygen from outside particles and/or from the entrapped oxygen inside the particles to cross the wall barriers. Drapala et al. [14] and Ahsaei et al. [15] both suggested that to obtain good encapsulation efficiency and improve the physicochemical properties of core materials, spray-drying processing conditions, and especially feeding characteristics, play an important role. Meanwhile, Bakry et al. [16] investigated that wall materials and core-to-wall ratio are very important for the efficient encapsulation of flavor. The core materials must possess the following properties: good emulsifiers, low viscosity at high concentrations, good dissolution, network-forming properties, and the capability to control the flavor release.

Some lack of consensus does exist within previous studies. Sheu et al. found that the retention of ethyl caprylate during spray-drying was improved by reducing the emulsion size [17]. Soottiantawat et al. found that for the encapsulation of d-limonene, ethyl butyrate, and ethyl propionate, increasing the emulsion droplet size decreased the retention of flavors [18]. Risch et al. found that a smaller emulsion size yielded a larger flavor retention during encapsulation, with an emulsion size in the range of 0.9 mm and 4.0 mm; however, a shorter shelf-life was observed [19]. Additionally, all of the previous studies limited the particles size of spray-dried capsules at microcapsules (1–1000 μm). The encapsulation behavior and oxidative stability of nano-capsules (1–1000 nm) during storage have been rarely reported. In this study, SOEO micro-nanocapsules were first prepared by high-pressure microfluidization. Then, the produced emulsions were spray-dried to obtain the micro-encapsulation. Further encapsulation behavior (loading capacity, encapsulation efficiency, surface oil, powder morphology, and *Tg*), release kinetics, and oxidative stability during accelerated storage were investigated. This study will help the industry to design appropriately sized capsule particles that will provide better retention and less oxidation of flavors during spray drying and further storage.

## 2. Materials and Methods

### 2.1. Materials

Sweet orange oil was provided by Guangzhou Baihua Flavours and Fragrances Co. Ltd. (Guangzhou, China). Ester gum was purchased from Wuhan Yuancheng Gongchuang Technology Co. Ltd. (Wuhan, China). Modified starch Capsul^®^ and maltodextrin with 18 DE were purchased from Ingredion Co. Ltd. (Shanghai, China) and Henan feitian Co. Ltd. (Hebi, China), respectively. Other chemicals were of reagent grade.

### 2.2. Emulsion Preparation

Modified food starch was added to deionized water to obtain a 20% (*w*/*w*) solids suspension (0.64 kg modified starch and 3.2 kg distilled water), stirring at 80 °C at room temperature with a magnetic stirrer for 2 h to be fully dissolved and then placed at room temperature overnight to hydrate. The dispersed phase consisted of 0.32 kg sweet orange oil and ester gum with a mass ratio of 3:2. The dispersed phase was slowly and evenly added to the carrier phase at a ratio of 1:2 at room temperature during homogenization by T25 Basic ULTRA-TURRAX homogenizer (IKA-Werke Co., Ltd., Staufen, Germany) at 9500 rpm. After being fully mixed, the mixture was further homogenized for 5 min to obtain the coarse emulsion. Then, the coarse emulsion was divided into three parts and immediately emulsified. The emulsions with different particle sizes were made by controlling the homogenization pressure and the numbers of passes. Two parts of the coarse emulsion were then subjected to a high-pressure microfluidic system (Nano DeBEE Co., Ltd., South Easton, MA, USA) at various pressures (5000 and 22,000 psi) for two and three passes, respectively. The emulsions after different pressure treatments were collected. Then, a maltodextrin of 373.33 g was added individually to each of the three different emulsions, respectively, and mixed for 10 min by a blade mixer (IKA-Werke Co., Ltd., Staufen, Germany). The individual emulsions were collected and marked as: large-size emulsion (LE), small-size emulsion (SE) and nano-emulsion (NE). The final emulsion contained 11.3% (*w*/*w*) modified starch, 5.7% (*w*/*w*) mixed oil phase, 26.4% (*w*/*w*) maltodextrin and 56.6% (*w*/*w*) distilled water.

### 2.3. Microencapsulation by Spray Drying

A total of about 1.41 kg for each emulsion were spray dried with an SD-1500 spray drier (Wodi Test Technology Co., Ltd., Shanghai, China). The spray drying operation conditions were as follows: inlet air temperature 150 °C, outlet air temperature 60 °C, fan speed 50.0 Hz, peristaltic pump speed 1.1 L/h. Three different particle sizes of encapsulated orange oil powder were obtained and marked as: large-size powder (LP), small-size powder (SP), and nano-size powder (NP). The collected LP, SP, and NP were placed in a container before wrapping with aluminum foil to avoid the light, and then stored in a refrigerator.

### 2.4. Feed Emulsion Droplet Size Analysis

The droplet size of the emulsions was determined using an LA-950 laser scattering particle analyzer (Horiba Co., Ltd., Kyoto, Japan). Distilled water was used as the dispersant. Approximately 50–500 µL of each emulsion was added to the instrument and the droplet size distribution was expressed as per the following Equation (1):(1)$$D_{43}=\frac{\sum_{i}z_{i}D_{i}^{4}}{\sum_{i}z_{i}D_{i}^{3}}$$
where D_43_ is the average emulsion size, z_i_ is the number of droplets of diameter D_i_. Each sample was determined twice, and the average was taken.

### 2.5. Reconstituted Emulsion Droplet Size Analysis

The droplet size distribution of the encapsulated powder was determined by stirring 1 g powder with 9 g of distilled water at room temperature for 5 min [20]. Then, the particle size distribution of the reconstituted emulsion was determined as described above.

### 2.6. Total Oil Content Determination

The total oil content in the microcapsules was determined following an early report [20]. A total of 20 g of spray-dried powder was dissolved in 200 mL of distilled water. The solution was distilled for 3 h, and the volume of distilled oil was determined. The result was presented in grams of oil, multiplying by the density of the oil (0.84 g/mL).

### 2.7. Surface Oil Content Determination

The surface oil content of encapsulated powders was determined as described by Finney et al. [21]. A total of 10 g of powder, 50 mL of hexane, and 1 mL of internal standard solution (3 mg/g 2-heptanone in acetone) were placed in a volumetric flask and stirred for 10 min to extract the oil. The extract was then filtered and the filtrate was recovered for further drying down with N_2_ until 1 mL of solution was left for determination by gas chromatography–mass spectrometer (GC-MS). Each sample was analyzed in triplicate. The content of surface oil in the powders was quantified by the following Equation (2):(2)$$\mathrm{Lim}=100\times \frac{{\mathrm{IS}}_{\mathrm{am}}}{{\mathrm{IS}}_{\mathrm{area}}}\times \mathrm{ArLim}\;\times \;\mathrm{RRF}$$
where Lim is surface oil amount, IS_am_ is the internal standard amount, IS_area_ is the internal standard peak area, ArLim is the d-limonene peak area, and RRF is the relative ratio of orange oil to d-limonene. Surface oil was expressed as g of surface oil per 100 g of encapsulated powder.

Encapsulation efficiency (EE) was calculated by the Equation (3), below, and the loading capacity (LC) was calculated by Equation (4), [22,23] below:(3)$$\mathrm{EE}(\%)=(\frac{\mathrm{Total}\;\mathrm{oil}\;(\frac{g}{g})\;\mathrm{powder}-\;\mathrm{surface}\;\mathrm{oil}\;(\frac{g}{g})\mathrm{powder}}{\mathrm{Total}\;\mathrm{oil}\;(\frac{g}{g})\mathrm{powder}})\times 100$$
(4)$$\mathrm{LC}(\%)=\frac{\mathrm{Total}\;\mathrm{oil}\;\mathrm{experimental}\;\mathrm{loading}\;(\frac{g}{g})\mathrm{powder}}{\mathrm{theoretical}\;\mathrm{loading}\;(\frac{g}{g})\mathrm{powder}}\times 100$$

### 2.8. Water Activity and Moisture Content Determination

The water activity of the powder was measured with an AW-2A water activity meter (Wuxi Bibo Corporation, Wuxi, China) at 25 °C. The moisture content was determined following the Official Methods of Analysis (18th ed.) [24].

### 2.9. Storage Stability of Encapsulated Powders

The storage stability of spray-dried powders of different particle sizes was expressed as the release and oxidation of d-limonene when stored at different water activities [25]. Desiccators with a_w_ of 0.25–0.75 were adjusted by saturated salt solutions of CH_3_COOK, MgCl_2_, Mg(NO_3_)_2_ and NaCl, respectively (GB5009. 238-2016). After spray drying, about 20 g of the spray-dried powder was spread in a 15 mL (20^Φ^ × 48 mm) glass bottle, and placed in a 37 °C desiccator. Samples were taken at 0 (after sample equilibration), 1, 2, 3, 4, and 5 weeks, and the d-limonene and limonene oxide were determined by GC-MS.

#### 2.9.1. Statistical Analysis

The release kinetics of d-limonene in the encapsulated powder were quantified by the Avrami Equation (5), below [26,27]:(5)$$R=\exp\left[-{\left(\mathrm{kt}\right)}^{n}\right]$$
where R is the retention of d-limonene, *n* is the release mechanism parameter, k is the release rate constant, and *t* is the storage time. The amount of limonene oxide expressed as mg of the oxide per g of the retained d-limonene was calculated by Equation (6):(6)$$C_{\mathrm{lo}}=\frac{W_{t}}{W_{T}}\times 100$$
where C_lo_ is the amount of limonene oxide, W*_t_* is the peak area of limonene oxide after *t* week storage, W_T_ is the peak area of d-limonene after T week storage, and *t* (T) is the storage time. When the limonene oxide content was higher than 2 mg/g d-limonene, it indicates that the shelf life of the powder had expired [26].

Furthermore, since the amount of limonene oxide increased linearly with time during the initial period, the apparent oxidation rate constants were calculated on the basis of the zero order reaction kinetic [21] with the following Equation (7):A = A_0_ + K_0_*t*(7)
where *t* is the storage time, A_0_ is the content of limonene oxide produced after the powder is equilibrated with different water activity, mg/g d-limonene, A is the content of limonene oxide at the *t* week of storage, mg/g d-limonene, K_0_ is the zero-order reaction rate constant, t^−1^.

All of the results are the average of three determinations unless otherwise stated. The results are shown as the average ± standard deviation (SD). Statistical differences were determined by *ANOVA* using Tukey’s post-hoc test (Minitab 19.0, State College, PA, USA). A significant difference between the average was defined as *p* < 0.05.

#### 2.9.2. GC-MS Analysis

GC-MS was used to determine d-limonene and limonene oxide [28]. Powders of 0.15 g were dispersed into 0.85 g of distilled water and mixed with a vortex mixer until completely dissolved. Then, 4 mL of internal standard solution (3 mg/g 2-heptanone dissolved in acetone) was added slowly and mixed for 5 min. The mixed solution was allowed to settle for 1 h. Next, the supernatant was filtered into the injection vial. Analyses were made in 7890A-5975C gas chromatograph (Agilent Co., Ltd., Santa Clara, CA, USA) under the following conditions: column DB-1MS (30 m × 0.25 mm × 0.25 μm, Agilent, Santa Clara, CA, USA); carrier gas helium; splitless injection; initial temperature 50 °C; initial time 2 min; program rate: 10 °C/min to 140 °C and holding for 2 min, 30 °C/min to 220 °C and holding for 2 min; injection volume 1 μL.

#### 2.9.3. Morphology by Scanning Electron Microscopy (SEM)

The microcapsules were observed by SEM (EVO-MA15, ZEISS Corporation, Göttingen, Germany) operating at 15 kV. The samples were fixed to a copper conductive tape with double glue, which was covered with a layer of coal (20 nm thickness). The samples were then coated with a thin layer of gold (50 nm) using a desk sputter coater DST1 system with a magnetron cathode under vacuum. The external structure was investigated as described by Carneiro et al. [29].

#### 2.9.4. Determination of Glass Transition Temperature (*Tg*) by Differential Scanning Calorimetry (DSC)

The glass transition temperature of the samples was obtained using a DSC-5000 differential scanning calorimeter (METTLER TOLEDO Corporation, Columbus, OH, USA). Approximately 10 mg of each spray dried powder with different water activity was hermetically sealed in an aluminum crucible. An empty aluminum pan was used as a reference. The heated rates were 10 °C/min and the scanning temperature ranged between 0 °C to 200 °C. The cooled rates were 10 °C/min and the scanning temperature was −20 °C [30]. The *Tg* was determined at the inflection point on the second heat ramp of the thermograph.

## 3. Results and Discussion

### 3.1. The Physical Properties of the Spray-Dried Powder

Three types of spray-dried powders of different particle sizes were prepared from emulsions with three different emulsion sizes. It has been reported that microfluidizer can make fine emulsions for different applications. Ozturk and Turasan [31] showed that emulsions made by microfluidization could produce emulsions with much smaller droplet sizes and narrower size distribution than those prepared from traditional homogenizer. In this study, different microfluidic pressures and numbers of passes were applied to make emulsions with different particle sizes.

The physical properties of the spray-dried powders, such as the feed emulsion size, reconstituted emulsion size, total oil content, surface oil content, loading capacity, encapsulation efficiency, water content, and water activity were shown in Table 1. From the results, it was not surprising that the droplet size of emulsions was decreased with the increasing of pressure and the droplet size distribution tends to be narrower and more uniform, as shown in the Figure 1. The reduction values of droplet size were 93% and 94% after the treatments of 5000 and 22,000 psi. After three passes under 22,000 psi, the smallest droplet size was reduced to 200 nm, which may be due to the accelerated adsorption to the oil/water interface because of hydrophobic groups on the hydrolyzed starch molecules. Zhang and Reineccius [32] found that the droplet size of an orange oil emulsion with modified starch was 169 nm at 22,000 psi for three passes and the droplet size would be further decreased by increasing the microfluidic pressure and numbers of passes. Chen and Wagner [33] reported that when pressure was increased from 2900 to 43,500 psi, a vitamin E nanoemulsion was produced with modified starch as an emulsifier. However, Jafari and Mcclements [34] showed that that with modified waxy corn starch and Hi-Cap as emulsifiers, the droplet size of d-limonene emulsions increased from 160 to 215 nm when the microfluidic pressure increased from 3045 to 12,200 psi with one pass. The inconsistency of these results could be ascribed to the different modified starch types, the emulsified compounds, and the machine of the high-pressure microfluidic system used. It has been recognized that high pressures can decrease the emulsion droplet size due to the magnitude of the disruptive forces generated by the microfluidic chamber. However, because of the limiting amount of emulsifier in the system and physicochemical modification of the emulsifier under high pressure, the droplet size might not continue to decrease with the increasing high pressure [34]. After the high-pressure treatment, the viscosity of the emulsion was decreased, the particle velocity was accelerated, and small particles were observed to easily aggregate into large droplets. So, a few large particles have been observed using a pressure of 22,000 psi as shown in the Figure 1C.

The droplet sizes of the reconstituted emulsions were as follows: LP, 1.17 μm; SP, 0.31 μm; and NP, 0.17 μm (Table 1). The particle size distribution of the LP after spray drying exhibited a bimodal distribution as shown in Figure 1A and droplet size was significantly decreased about 65% after spray drying. The change of particle size during the spray drying may also be related to the speed of the atomizer and the size of the nozzle [35]. Except for the LE, there was no significant different in droplet size between feed emulsions and reconstituted emulsions for the other two emulsions, and both the emulsions and the spray-dried powder demonstrated monomodal size distribution. To maximize volatile retention and minimize oxidation during the spray drying, the feed emulsion size and the reconstituted emulsion size should be similar to prevent the exposure of the active ingredient under the high temperatures of the spray drier.

It was observed that the total oil content and loading capacity were generally decreased with the reduction of the particle size, as shown in Table 1. The highest total oil content and loading capacity was in the LP and the lowest was in the NP. The total oil content in the LP and NP was 1.39 and 0.93 g/20 g powder, respectively. The loading capacity in the LP and NP was 99.6% and 77.7%, respectively. The reason for the lowest total oil content and loading capacity of the NP might be flavor loss during the homogenization with an increase of pressure and numbers of passes since there were accelerated frictions between the material and the homogenization valve. Our result was in accordance with Linke et al. [36], who found that the large atomized particles have smaller surface area to volume ratios and these type of particles could contributed to high orange oil retention. As shown in Table 1, higher surface oil content was observed for the LP. It was found that with the decrease of the emulsion size, the surface oil content was decreased. This finding was in accordance with earlier reports [18]. In addition, except for the higher surface oil content, the lowest encapsulation efficiency was found in the LP, although the encapsulation efficiency was higher than 95% for the three different types of powders. The lower surface oil content and higher encapsulation efficiency had an important effect on the storage stability of flavor in the spray-dried powder [37]. The water content of the powders of different particle sizes was less than 4%, which was in the range of moisture content of commercial products after spray drying (3–6%) [38]. The water activity of spray-dried powder was in the range of 0.24~0.30 and SP had the highest a_w_ (0.33). Finney, Buffo, and Reineccius [21] reported that the particle size had no significant effect on the water content and water activity of the powder, and that the inlet and outlet air temperature during spray drying should be a critical factor.

### 3.2. Morphological Characterization by Scanning Electron Microscopy (SEM)

After spray-drying, the three types of powder were immediately subjected to SEM and the outer structure was shown in Figure 2. The SEM images of all powders showed similar morphological characteristics including spheroid shapes without shriveled surfaces, and fissures or cracks in the structure. Those microphotographs of varied particle sizes were captured under different magnification times and presented both groove and smooth surfaces.

### 3.3. Release Kinetics of D-Limonene from Spray-Dried Powders during Storage

The release time profiles of d-limonene in spray-dried powders were measured for 5 weeks under accelerated conditions (37 °C) and under various a_w_ (0.25–0.75) as shown in Figure 3. After 5 weeks of storage, the retention rate (R) of all three types of powder decreased to varying degrees. During storage, the highest R values for d-limonene were observed for the lowest a_w_ (0.25), and Rs of 95.6%, 97.4%, and 96.1% were observed for LP, SP, and NP, respectively. When the a_w_ increased from 0.25 to 0.75, the R decreased to 71.3% and 83.4% for LP and SP, while NP still contained 91.9% d-limonene. This result indicates that both particle size and a_w_ have a significant impact on the retention of flavor compounds. The retention rate correlated to the properties of the wall material, the outer surface morphology, as well as the oxidation of d-limonene. Once d-limonene is oxidized, the R decreases. The microstructure of the capsule allows oxygen and water to penetrate the wall material and enter the core, causing the d-limonene to release or oxidize. Especially when stored under high a_w_, more water is adsorbed in the wall material, hydrating the powder. This causes the powder to swell, causing the collapse of the outer surface morphology, and producing fissures on the surface, especially for large particle size microcapsules that have more fissures or will crack more easily, which leads to the easier release or oxidation of the flavors. In contrast, the outer surfaces of capsules containing nano-particles tend to be more complete, smooth, spherical, and without dents and shriveling. Furthermore, the d-limonene in the nano-sized powder is less likely to be oxidized, achieving a high retention rate.

From Figure 3, it can be seen that the release kinetics of d-limonene were correlated well with the Avrami equation and the release rate constant (k) and release mechanism parameter (n) against the reconstituted emulsion size were calculated and shown in Figure 1. For the release rate constant, k, with the particle size decreasing, k gradually decreased, and the NP had the lowest release constant. Water activity also showed a pronounced effect on the k of d-limonene. At low aw, only a small amount of d-limonene was released. As the water activity increased from 0.25 to 0.33, the k of the d-limonene in each three powders dramatically increased by 5.1-, 6.4- and 1.2-fold. Generally, at low aw, the encapsulated particles retain their original shape and the wall materials remains in a glassy state, still acting as a good barrier against the passage of water, oxygen and d-limonene. As the water activity continued to increase to 0.33, the capsule matrices began to be plasticized, and with these structural changes, the very high mobility of d-limonene has been observed. From 0.52 to 0.75, k of LP, SP and NP was increased slowly and 37%, 19% and 8% have been increased, respectively. The slow increase of k values at high humidity could be attributed to the reduced release of d-limonene when stored at high water activity (0.52~0.75), as the effective surface area of the carrier is reduced and the encapsulated powder begins to rehydrate. Most particles may clump and stick together, turning into a paste with a rubbery capsule matrix.

To further investigate how powder morphology impacts flavor release, surface area, surface porosity, and surface integrity were observed under SEM after storage for 5 weeks at 37 °C/0.33 a_w_ as shown in Figure 4. The particles of LP and SP were detected to swell, clump, and adhere together, which may explain the high release of LP and SP at this water activity level. In contrast, the NP retained its original shape with good dispersion and better protection of d-limonene, as shown in Figure 4C.

As previously reported by Yoshii, et al. [26], the *n* of the Avrami equation was related to the release mechanism. Theoretically, the release equation for d-limonene should follow zero-order kinetics; *n* should be zero and the release rate of flavor should be constant. However, due to several factors, the release kinetics are not always of zero order. As shown in Figure 1b, at the lower a_w_ values of 0.25 and 0.33, the *n* of the three powders is below one, indicating that the molecular diffusion rate of d-limonene was limited, with only SP having an *n* of 0.99 at a_w_ values of 0.33, representing a first-order reaction release mechanism. At high a_w_ (0.75), *n* was greater than one, indicating that the increase in water activity has a significant effect on the release of d-limonene.

### 3.4. Glass Transition Temperature

The release pattern of encapsulated d-limonene at different water activities indicates structural changes in the carrier, which may reflect changes in the glass transition temperature, *Tg. Tg* was used as the phase transition index. Therefore, the effects of a_w_ on the *Tg* of the encapsulation carrier were measured at 37 °C as shown in Table 2. At low a_w_ values, the *Tg* of LP, SP, and NP were significant different, at 62, 88, and 100 °C, respectively, with NP > SP > LP. This indicates that the smaller the particle size, the higher the *Tg*. The *Tg* of the three powders was much higher than the storage temperature (37 °C), at which stage the powders were in a glassy state and the release of d-limonene was limited and *n* < 1. Additionally, NP had the highest *Tg*, which was also in accordance with the lowest release rate (k) and n. However, as the a_w_ increases to 0.75, the *Tg* decreases to 42 °C, which is close to 37 °C. At this stage, the powders are in the rubbery state and the three powders have very high k and n.

### 3.5. Oxidative Stability of D-Limonene in Spray-Dried Powders

The stability of orange oil can be demonstrated by the release of d-limonene from spray-dried particles and the production of oxidized compounds such as limonene oxide (limonene-1, 2-epoxide). The time course of limonene oxide formation in powders was studied at 37 °C and 0.25–0.75 a_w_ in Figure 5 and Figure 6. The formation of oxides was strongly related to a_w_, particle size, and storage time. As shown in Figure 5, the three powders had about 0.6~1.0 mg limonene oxide/(g d-limonene) at 0 weeks. However, at 5 weeks, although the oxidation of d-limonene in NP was much slower than in LP and SP, they increased linearly with increasing storage time. The formation of oxides reached a maximum at the a_w_ of 0.33, where it was the lowest in NP (0.26 mg/g d-limonene/week) and SP and LP demonstrated 1.14 and 1.08 mg/g d-limonene/week, respectively. The formation of oxides in the NP was 3.4-fold lower than for the LP and SP. The most likely reason was that the reduction of specific surface area in NP reduced the migration of oxygen and moisture. In contrast, the matrix structures of the LP and SP are significantly adherent, which may lead to an increase in intermolecular voids and the mobility of oxygen, as shown in Figure 4A,C.

The apparent oxidation rate constant was further calculated according to the zero-order kinetic, Equation (7), which markedly depended upon water activity, as shown in Figure 6. For the LP and SP, the formation of oxides was highest at an a_w_ value of 0.33 and then decreased at the higher a_w_ values of 0.52 and 0.75 during storage. There are some possible reasons to explain these results. Limonene oxides are further degraded to form other oxide compounds with smaller molecular weights, which may be easily released into the surrounding environment. Additionally the oxygen concentration may decrease at high a_w_ values because of the low solubility of oxygen in water, which reduces the oxidation [25].

## 4. Conclusions

The encapsulation characteristics, release behavior, and d-limonene oxides of encapsulated orange oil with different particle sizes were investigated under different a_w_ values at 37 °C. The particle size and water activity had a significant effect on the storage stability and release of d-limonene in spray-dried powders. At different a_w_ values, the retention rates were always higher and the release and oxidation rates were always lower than those of LP and SP. With the increase of water activity (0.25 to 0.75), the release rates of LP, SP, and NP first increased rapidly until reaching the a_w_ of 0.33, and then increased slowly, while the oxidation rates first increased rapidly to an a_w_ value of 0.33 and then decreased. An a_w_ value of 0.33 was critical for the release and oxidation of d-limonene from the three powders, with the largest release and oxidation rates of 3.76 × 10^2^/week and 1.14 mg/g d-limonene/week for the larger sized powders, LP and SP, respectively, in contrast to the lowest release rates of 1.02 × 10^2^/week and 0.26 mg/g d-limonene/week for the NP. The release characteristics and oxidation stability of d-limonene were strongly influenced by the powder morphology and glass transition temperature of the spray-dried powders. The more complete the surface morphology of the powder, the higher the storage stability.

## Data Availability

All related data and methods are presented in this paper. Additional inquiries should be addressed to the corresponding author.

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
