# Peer review of "Effect of Emulsion Particle Size on the Encapsulation Behavior and Oxidative Stability of Spray Microencapsulated Sweet Orange Oil (Citrus aurantium var. dulcis)"

_foods, 2022, doi:10.3390/foods12010116_

Round 1

Reviewer 1 Report

In this manuscript, authors developed microencapsulation process for sweet orange oil. Authors formulated three formulations of microcapsules (large, small and nano sized) and studied the physicochemical characteristics of the microcapsules. The manuscript was written well with detailed analysis. Following are my queries and suggestion: 

Authors developed three emulsion formulation (large-size emulsion, small-size emulsion and nano-emulsion) varying significantly in its particle size. To maintain stability, the requirement of emulsifier should differs for each formulations. For instance, the nanosized formulations require more quantity of emulsifier than the emulsifier required for large-sized emulsion. However, in this study, authors used the same quantity of emulsifier for the three formulations. Authors can explain why this difference in emulsifier requirement was not considered?

Line 65: “Based on the previous studies, there are some controversy”. A brief discussion on the controversy with the previous work will be interesting to read.

Line 68: “High-pressure microfluidization combined with spray drying”. This statement conveys, the outlet of the microfluidization process will be the inlet for atomizer in the spray dryer, which is not possible. Author can rephrase this sentence. 

Section 2.9.1 “Statistical Analysis”: In this section, only the release kinetics was described and the description on statistical analysis considered in this study was missing.

Line 218, 244: In the text authors mentioned, supplement 1 and 2. But it was not available to refer. Does authors mentioned as Scheme 1 and 2?

Section 3.2: Authors can provide the SEM images taken in for the group of particles, and can also provide a magnified image of selected particle. This will confirm the spray dried particles resulted without smooth surface, fissures or cracks. 

Reviewer 2 Report

The manuscript is good quality, but the calculation of the encapsualtion efficiency is not clear. Units in which the content of surface and total oil and, consequently EE are expressed can be confusing. 

I wonder how it is possible that since EE is closely related to the content of surface fat (formula no. 3), despite large differences in the content of surface oil, EE is still at a similar level for all samples. 

Round 2

Reviewer 1 Report

Authors responded the queries and modified the manuscript.